# OpenReview forum: "MORALISE: A Structured Benchmark for Moral Alignment in Visual Language Models"
_ICML.cc/2026/Conference — ICML 2026 regular_

### Official Review · Reviewer_N84j · 2026-03-06

**Soundness:** 3
**Presentation:** 4
**Significance:** 2
**Originality:** 1
**Overall Recommendation:** 4
**Confidence:** 4

**Summary:**

This paper develops a benchmark to evaluate moral understanding and alignment of multimodal large language models. To construct the benchmark, the paper develops a taxonomy which categorizes 13 moral topics into 3 overarching domains. For each domain, annotators collect suitable images and text-image pairs from online sources and have multiple annotators labeling them. Samples with high agreement are included in the benchmark.

Based on the collected samples, two tasks are defined: Moral Judgement, i.e. deciding whether the action shown in the image is "morally wrong" or "not morally wrong", and Moral Norm Attribution, i.e. deciding to which topic the action shown in the image belongs. The paper then evaluates a large number of open MLLMs and also GPT variants on the benchmark and analyzes trends, for example stronger performance on Moral Judgement than on Moral Norm Attribution, scaling trends (i.e. larger models performing better but then plateauing), and responses are more strongly correlated within model families than across them.

**Compliance With Llm Reviewing Policy:**

Affirmed.

**Final Justification:**

After rebuttal and discussion, arguments for acceptance outweigh arguments against acceptance. In light of the paper's merits, novelty aspect and annotator demographics are not strong enough arguments to justify rejection.

**Key Questions For Authors:**

For a revision, the paper should address the following questions:
 * Which insights are enabled through the proposed benchmark that we cannot gain from evaluating the same set of models on  M$^3$oralBench?
 * What is the "performance" of external, untrained human annotators on the Moral Judgement and Moral Norm Attribution tasks?
 * How can we obtain labels that state whether a behavior is "morally wrong" in a valid and generalizable way?

**Limitations:**

The paper includes two limitations that are mostly not discussed:
 * Who are the human annotators that performed labeling? Were they authors or externally hired workers? If so, what was their compensation?
 * What are the conditions to collect labels for "morally wrong" behaviors in a valid way?

**Strengths And Weaknesses:**

## Strengths
**(S1)** The benchmark addresses a timely and important topic, i.e. moral alignment of multimodal large language models

**(S2)** The paper evaluates a large and comprehensive set of models and analyzes a number of trends emerging from the results

**(S3)** Benchmark construction focuses on high inter-annotator agreement

**(S4)** The benchmark taxonomy is grounded in existing and recognized literature on human moral systems

**(S5)** The writing and figures are overall clear and the supplementary material provides details to reproduce experiments, e.g. prompts

---
## Weaknesses
### Major Weaknesses
**(W1)** The proposed benchmark is very similar to M$^3$oralBench [1]. For example, both benchmarks share the tasks of Moral Judgement and Moral Norm Attribution. Both papers include the cross-model correlation analysis. The present paper mainly differs on (1) curating natural images (vs. synthetic images in M$^3$oralBench); (2) choosing a different taxonomy of moral topics and different literature to ground it in; and (3) evaluating more models as well as providing more fine-grained analysis about trends. However, given the close similarities, the paper has to provide evidence or discussion regarding the following:
  * Are there any concrete, quantifiable problems with using synthetic images? The paper claims several times (e.g. l. 033, l. 084) that using AI images for benchmarking Moral Alignment could be problematic, but does not provide any evidence for this claim.
  * Could the same results (e.g. regarding scaling trends, within-family similarities, different performance of Moral Judgement and Moral Norm Attribution, ...) be obtained with evaluating the same set of MLLMs chosen in the present paper by evaluating them on  M$^3$oralBench?
  * Why did the present paper choose a different taxonomy of moral topics than  M$^3$oralBench? What are relevant differences that make the one in the present paper better suited for analysis?

In summary, the paper needs to provide stronger, and also empirical, arguments for the need of a new Moral Alignment benchmark, given that  M$^3$oralBench already exists and is very similar in structure to the proposed benchmark.

**(W2)** `Takeaway 1` and `Takeaway 2` rely on the assumption that the samples (images or image-text pairs) in the benchmark actually contain enough unambiguous information to correctly identify the relavent Norms in Moral Norm Attribution. However, this was not tested. Annotations were provided by trained annotators who are familiar with the taxonomy and have seen many examples of scenarios such as those included in the benchmark. The paper does not include a study with external participants who are given only the same information (i.e. prompt) that was provided to the MLLM models. However, this evaluation is necessary to understand the reasonable upper bound performance of the benchmark.

**(W3)** The benchmark construction is missing critical details, such as how the texts for image-text pairs are created (i.e. whether annotators formulate them or they are alt-text captions). Also, the paper does not contain information regarding how it was decided to include an image as standalone sample or together with paired text. Finally, there are no qunatitative analyses regarding the diversity (only one informal statement in l. 201). Good scientific practice would also recommend to specify whether annotators are authors of the paper (which I think could be problematic), how many annotators there are, and how much they were paid for their work.

**(W4)** There is some ambiguity in the paper regarding the normative status of annotations. The paper refers to "morally wrong" and "not morally wrong" samples, but Fig. 7 (in the legend) calls them "wrong" and "correct" (which is not the same as "not morally wrong"). At the same time, the caption of Fig. 7 refers to "morally neutral" samples, which again has a different nuance. Since this aspect is central to the benchmark and the paper, careful conceptualization is required. This also points to a fundamental issue that is not sufficiently discussed: But classifying samples, i.e. scenarios, as "morally wrong" the benchmarks makes normative statements about the moral value of certain behaviors. But it does not include a discussion on how such labels could be justified. The pragmatic approach taken by the paper, i.e. "$k$ annotators agree it is morally wrong, so the label is justified", has limitations, for example it could be argued that some moral judgements are independent of majority vote, while in other cases the background of annotators (graduate students in this case) could differ significantly from the general population. At least, the paper should discuss in detail what are problems with creating and releasing a benchmark that makes normative statements about the moral value of the included scenarios, and discuss procedures to obtain such labels in a valid way.

---
### Minor Weaknesses
**(W5)** The paper does not explicitly specify whether code and data will be released. Perhaps l. 099 implies this, but an explicit confirmation would help.

**(W6)** While writing and figures are generally of high quality, some minor aspects such as the widows (hanging lines) in line 178 and line 062 (right column) could be improved.

**(W7)** The paper does not contain any actionable insights on how to improve models for Moral Alignment. This argument alone is clearly not grounds for rejection and should not be required of a benchmarking paper, but if the authors have insights on how to improve models, that would be appreciated. On a related note, in line 197 and Sec. 5 Conclusions the paper mentions "debias the moral reasoning capabilities". It is unclear what this refers to, as "bias" was not discussed otherwise in the paper. I.e. what does "bias" refer to in the context of Moral Alignment?

### References
[1] Yan, Bei, et al. "M$^3$oralBench: A MultiModal Moral Benchmark for LVLMs." arXiv preprint arXiv:2412.20718 (2024).

---

> ### Author Rebuttal · Authors · 2026-03-31
>
> We sincerely thank you for the careful and constructive review. Below are our responses to your points.
>
> ## W1: Difference with M3oralBench
> 1. **Problems with synthetic images.**
> Prior and recent studies show that synthetic images can look strong but still shift model predictions in unexpected ways [1,2]. AI-generated images also frequently has flaws (e.g., distortion in background is visible even in the M3B main example in Fig. 5). We cannot yet quantify how much image-quality issues change results, as that requires matched real/synthetic version of M3B/Moralise. But we believe synthetic images should not be a drop-in replacement for real-world evaluation data, given potential artifacts and hidden biases [3,4]. **Our position is that natural-image (such as Moralise) is at least a necessary complement, and we will state this explicitly in the paper.**
> 2. **Can the same conclusions be derived from M3B?**
> Partially. We ran corresponding experiments on M3B to the best extent possible, models >14B are excluded due to resource constraints. **Some general findings should transfer, and in our runs they do**: such as model-family similarities, the judgment-vs-attribution performance gap. For scaling trends, we noticed differences in family-specific behaviors. **Conclusions that depend on benchmark design are not derivable from M3B**, such as topic-level difficulty (Moralise has a finer taxonomy) and the comparison of image-centric vs. text-centric settings (M3B lacks this dimension).
> 3. **Why a new taxonomy?**
> **Our main motivation is to decouple separable concepts:** e.g., fairness and cheating are grouped as one foudation in M3B.
> But there is a conceptual distinction between unfairness stemming from social prejudice and improper individual conduct. The latter falls under personal-integrity in Moralise. We will discuss this design choice in the revision.
>
> [1] Lance: Stress-testing visual models by generating language-guided counterfactual images. NIPS23
> [2] A-bench: Are lmms masters at evaluating ai-generated images? ICLR25
> [3] The unmet promise of synthetic training images: Using retrieved real images performs better. NIPS24
> [4] A review of synthetic image data and its use in computer vision. Journal of Imaging 2022
>
> ## W2: Human evaluation
> Thank you for pressing this point. We ran an external study with 8 non-author undergrad/graduate volunteers, using exactly the same inputs and instructions as the VLMs. Majority-vote human performance is 94.1% accuracy, which is clearly above best model performance. We tried the best to find volunteers from diverse ethnicities and cultural backgrounds to ensure unbiased results. Details on the study design and discussions will be included in the revision.
>
> ## W3: Annotation details
> 1. **For image-text items, text is created by annotators (not alt-text).**
> 2. **The standalone/paired is NOT decided post hoc per image. We use two pipelines by design:** *text-centric* (start from scenario, retrieve image) and *image-centric* (start from image, then annotate context). This naturally produces image-only and image-text samples.
> 3. **We have 12 annotators, 3 authors and 9 external PhD/PhD-track experts.** All recruited from main author's institutions as unpaid volunteers. We will clarify on these details and acknowledge contributors in revision.
>
> ## W4: Conceptualization and normative status
> Thank you for your careful reading!
> 1. We will standardize terminology to morally wrong vs. not morally wrong and make figure/caption wording fully consistent.
> 2. We fully acknowledge that the annotator team under our resource constraints inevitably has potential limitations, which is why we maintain high-frequency discussion within the annotator team to maximize awareness of potential bias.
> 3. Following your suggestion, we will note the operational nature of labels and include more quality-assurance details (e.g., annotator training and quality control). Also, alongside with dataset, we will setup a channel for public discussion and feedback on contested cases.
>
>
> ## W5: Artifacts
> Yes, our dataset and codebase are ready for release!
>
> ## W6: Writing improvements
> We appreciate your recognition of our writing quality! We will fix those issues.
>
> ## W7: Further discussions
> We will expand the discussion to actionable directions suggested by our results: e.g., (1) given the difficulty on uncommon ethic norms, collecting alignment data from underrepresented groups and topics is important; (2) as visual-centric moral reasoning remains difficult, alignment grounded in visual evidence likely needs dedicated treatment beyond direct transfer from text-only alignment, and Moralise can be an initial foundation for visual-centric alignment data. Finally in our paper, "bias" refers to distortion from desired moral reasoning behavior, which we will define explicitly and use consistently in the revision.
>
> **Hope our rebuttal addresses your concerns, and we are looking forward to further discussions!**

---

> > ### Author Rebuttal · Reviewer_N84j · 2026-03-31
> >
> > Thank you for the extensive, clear, and easy to follow response to my review. I strongly appreciate the author's honest take regarding the similarities to M3B, this was really refreshing to see.
> >
> > Below are my detailed responses to points worth further discussion:
> >
> > ---
> >
> > **(W1)** From my perspective, I have seen many benchmark papers either claiming synthetic images are the better choice, or natural images are the better choice, without providing evidence. My conclusion is that both can contain confounders or artifacts that skew evaluation. Therefore, I agree with the authors' argument that they should be treated as complementary. Hopefully, future benchmark construction can consider this and as a field we can proceed towards designing better benchmarks that include different domains and image curation methods. My suggestion is to stress the complementarity more than superiority to synthetic images in the paper.
> >
> > Regarding the similar behavior of models on M3B and MORALISE, thank you for clearly and honestly stating this. My suggestion is to include these results in the paper (in the supplementary material), and better explain differences in the underlying taxonomy, i.e. which aspects differ and why comparing these differences may be interesting.
> >
> > ---
> >
> > **(W3)** Could you please clarify how exactly you will standardize terminology regarding morally wrong vs. not morally wrong? I think explicitly aligning on this would be quite important.
> >
> > > Also, alongside with dataset, we will setup a channel for public discussion and feedback on contested cases.
> >
> > I think this is a very good solution, and I appreciate this effort.
> >
> > ---
> >
> > I assume other clarifications will be included as described.
> >
> > Overall, I now feel positive about the paper and important details have been provided in the rebuttal. As much as I appreciate the honesty and clarity regarding similarities to M3B, unfortunately, novelty remains an issue at a venue like ICML which certainly requires this for method papers. However, as I'm not strongly opposed to acceptance in light of the rebuttal and other reviewers also lean towards accept, I will attempt to clarify in the Reviewer-AC discussion whether there are different expectations for benchmark papers, and I will raise my overall rating and soundness score to 3 for now.

---

> > > ### Author Response · Authors · 2026-04-02
> > >
> > > **Dear Reviewer N84j,**
> > >
> > > **Very glad to hear that our rebuttal helped address your concerns. It is also rare to see a reviewer who initially held a negative opinion respond so promptly, objectively, and positively to our carefully prepared rebuttal :) We therefore want to express our sincere respect for the professionalism and responsibility you demonstrated as a reviewer.**
> > >
> > > Regarding the points you raised:
> > >
> > > 1. We are pleased that we reached consensus on this point. As you suggested, we will revise the wording in the paper to better emphasize the complementarity between synthetic and real image data. We also plan to add a new section, Further Experiments and Analysis, before the current Appendix C in the next version of the paper, in order to present the more substantial experiments conducted during the rebuttal period, including the results on M3B as well as the other related discussions you mentioned.
> > > 2. During annotation, we instructed annotators to label all samples describing scenes/behaviors/phenomena with potential moral risk (discrimination, harm, improper gain, etc) as “morally wrong,” and to label all other samples as “not morally wrong.” This means that “morally not wrong” does not refer only to morally commendable cases. It also includes morally neutral or morally irrelevant samples, such as an innocent scene like “a dog playing in the park,” which does not involve any moral judgment. We also thank you for recognizing our approach to handling contested cases.
> > >
> > > **Finally, thank you again for your quick response and for the corresponding score adjustment. Should you have any further questions, we would be very happy to discuss them.**

---

### Official Review · Reviewer_dr59 · 2026-03-12

**Soundness:** 3
**Presentation:** 3
**Significance:** 3
**Originality:** 3
**Overall Recommendation:** 4
**Confidence:** 4

**Summary:**

This paper introduces a new dataset for benchmarking the moral alignment of vision-language models called Moralise. A taxonomy based on Turiel’s Domain Theory is proposed to categorize the moral content of image-text examples into three domains: personal, interpersonal, and societal. The authors map 13 moral topic categories to these three domains to serve as a basis for constructing their dataset. The dataset itself consists of 2,481 image-text pairs which were manually collected from internet sources and validated by humans, with the explicit aim of avoiding AI-generated content. The authors evaluate 19 open weight and commercial VLMs using their dataset while also conducting several analyses such as the impact of model scale and differences in moral reasoning across modalities.

**Compliance With Llm Reviewing Policy:**

Affirmed.

**Final Justification:**

Most of my concerns have been addressed, with the exception of those which cannot be easily resolved during the rebuttal (all human annotators being college students & lack of additional examples for assessing dataset quality). Therefore, I have decided to maintain my initial scores which reflect my positive overall assessment of this paper.

**Key Questions For Authors:**

1. How many annotators evaluated each instance in Moralise?
2. Why did you choose 83% agreement as the minimum threshold for acceptance?
3. Line 272 mentions that greedy decoding was used in your experiments. However, I would assume that moral reasoning abilities are more relevant to generation settings in which sampling is used (e.g., creative writing and other open-ended generation tasks). Did you investigate how higher temperature sampling impacts moral judgment evaluations? More broadly, what are the generation contexts in which we would care about the model's ability to make moral judgments and in which greedy decoding would be an appropriate choice for sampling settings?
4. Given that GPT-o4-mini outperforms GPT-4o in moral reasoning, could this be due to the former being a reasoning model? Did you evaluate at all how simulated reasoning / long CoT impacts moral judgments?

**Limitations:**

The authors discuss limitations in Appendix E. I suggest adding further discussion on the limitation I described in W1 (all annotators being ML graduate students). It would also be worthwhile to add some discussion about how moral norms and intuitions are not universally agreed upon and may differ across races, cultures, etc.

**Strengths And Weaknesses:**

**Strengths**

1. This work addresses a timely and important problem considering the rapid adoption of VLMs in real-world settings and the relatively fewer prior studies examining moral reasoning in the vision domain (as compared to text).
2. The dataset is grounded in previous foundational work on moral reasoning taxonomies, providing credibility to its design and scope.
3. Human experts were employed to collect and annotate the dataset, which provides some assurance of its quality.
4. The experiments cover a wide range of model families and different model scales, which enables interesting analyses on the effect of model size and closed vs. open weight models.
5. Overall the paper is well-written and easy to follow

**Weaknesses**

1. The human annotators were all graduate students in an ML-related field. I suspect that individuals in this subgroup may not be representative of the broader population, which could have an effect on their moral judgments. It would be worthwhile to at least discuss the potential implications of this limitation in the paper.
2. Examples are categorized as being either text-centric or image-centric, which is nice because it enables analysis of moral reasoning abilities across different modalities (Section 4.4). However, in the case of image-centric examples, I wonder if there are still clues in the text descriptions which ultimately influence the moral judgments of the model more than the visual scenario. The image-centric examples in Figure 2 do not have corresponding text descriptions, which makes it impossible to judge whether or not this is a factor. One way of testing this is to provide only the image (without corresponding text description) for the image-centric examples.
3. Related to W2 described above, the only examples from the dataset are in Figure 2\. It would be helpful to provide additional examples in the appendix and provide the complete image-centric instances (including text descriptions) so that the reader can better assess the quality of the dataset. It would also be interesting to see examples where there was greater disagreement amongst the annotators as well as instances which have higher moral reasoning failure rates across models.
4. Some details of the annotation process are unclear. For example, lines 209-210 mention a majority vote protocol being used, but there is no mention of how many annotators evaluated each instance. Additionally, it’s unclear why 83% was chosen as the minimum threshold required for agreement among the annotators.
5. The main experimental results report only the average score and rank. I wonder how much variability exists in these results and if there are significant differences across models.

---

> ### Author Rebuttal · Authors · 2026-03-31
>
> ## W1. Annotator Bias
>
> Please kindly refer to our response to W2 of Reviewer UKrj.
>
> ## W2. Image-Centric Setting
>
> We wish to clarify that for image-centric samples, **we do NOT provide any image descriptions (i.e., the $<\text{text}>$ field is empty) and include only minimal task instructions** (see Appendix B.1 for prompt templates). As a result, model predictions are based solely on visual inputs, without any textual cues describing the image, i.e., our current setup already corresponds to the ablation study suggested by the reviewer. We will further clarify this point in paper to avoid potential confusion.
>
> ## W3. More Examples
>
> 1. The examples in Figure 2 are already complete instances, since image-centric instances do not include any text descriptions as clarified in W2.
>
> 2. we agree that providing more examples help assess data quality. In the revised version, we will include a broader set of examples in the appendix, along with their corresponding annotation agreement levels. Specifically, we will present samples with agreement levels of 12/12, 11/12, and 10/12, showing how agreement correlates with scenario clarity. (Agreement levels are explained in W4.)
>
> Since we cannot provide figures during rebuttal, we choose one representative image from three categories and provide their brief description below:
>
> - 12/12: A car runs a red light at an intersection.
> - 11/12: Factories emit thick black smoke, indicating severe air pollution.
> - 10/12: An elderly man lies on the floor, suggesting a possible injury.
>
> ## W4 (Q1, Q2). Annotation Protocol
>
> To ensure data quality, we adopt a structured pipeline: (1) 12 annotators independently collect examples from the web, with an explicit preference for clear moral violations rather than borderline cases. (2) Each instance is labeled by all 12 annotators. (3) We apply a majority-vote filtering: samples with fewer than 10 out of 12 annotators agreeing on the label are discarded. Therefore, the agreement threshold is 10/12 ≈ 83%. We will clarify those details in the revised version.
>
> ## W5. Clarification on Results
>
> We wish to clarify that **"average scores and ranks" only refers to the 2 columns on the right and 1 row at the bottom of the main result tables** (Table 2-4), which are computed based on the detailed numbers reported in those tables, respectively. **We provide fine-grained results across multiple dimensions, including tasks (e.g., judgment vs. attribution), models, modalities, and moral topics**.  In the appendix, we report additional breakdowns at the modality level. These detailed results explicitly reveal substantial variability induced by factors such as model architecture, input modality, and topic category, as summarized in relevant conclusions (e.g., Takeaways 3, 4, 5, 7).
>
> ## Q3. Decoding Settings
>
> For a benchmark setting, we consider fairness, determinism, and reproducibility as primary concerns. **We therefore adopt greedy decoding for all models, as it produces deterministic outputs and ensures stable and directly reproducible comparisons across runs.** In contrast, temperature-based sampling introduces randomness, which can increase evaluation variance, particularly for morally subtle or ambiguous cases where small generation differences may lead to different judgments. In such scenarios, the observed scores may reflect sampling noise rather than the model’s underlying moral reasoning ability. This setting is also consistent with prior work [1,2], and we thus use greedy decoding as a controlled evaluation protocol.
>
> More broadly, we agree that analyzing how temperature affects moral judgment is an important direction. However, this is not the main scope of our paper, and we will highlight it as future work.
>
> [1] Do VLMs Have a Moral Backbone? A Study on the Fragile Morality of Vision-Language Models
>
> [2] Moral Sycophancy in Vision Language Models
>
> ## Q4. Evaluation of Reasoning Models
>
> We do not attribute the stronger performance of o4-mini over GPT-4o to explicit reasoning traces, as neither model is allowed to use CoT in our experiments. Under this controlled setting, the performance difference should be understood as a model-level difference rather than an effect of exposed reasoning.
>
> To further examine this question, we conduct an additional experiment in which o4-mini is required to generate CoT. Compared with the no-CoT setting, the results show a mixed effect: CoT improves image-centric attribution (from 63.57% to 65.06%) but reduces text-centric attribution (from 80.90% to 77.84%). This suggests that **explicit reasoning may benefit certain modality-specific tasks, but does not yield consistent gains overall**. The underlying mechanism appears non-trivial, and understanding how long-form reasoning interacts with moral judgment remains an important direction for future work. We will include these results and discussion in the revised version.
>
> **Thank you for the constructive feedbacks. We look forward to further discussion.**

---

> > ### Author Rebuttal · Reviewer_dr59 · 2026-04-03
> >
> > I appreciate the authors' detailed response to my review. Most of my concerns have been addressed, with the exception of those which cannot be easily resolved during the rebuttal (all human annotators being college students & lack of additional examples for assessing dataset quality). Therefore, I have decided to maintain my initial scores which reflect my positive overall assessment of this paper.

---

### Official Review · Reviewer_UKrj · 2026-03-13

**Soundness:** 2
**Presentation:** 3
**Significance:** 2
**Originality:** 2
**Overall Recommendation:** 5
**Confidence:** 3

**Summary:**

This paper introduces MORALISE, a multimodal benchmark for evaluating the ability of VLM to assess moral content in image-text pairs. The authors apply this benchmark to characterize the moral behavior of both open- and closed-source VLMs across a range of moral topics and two evaluation tasks.

**Compliance With Llm Reviewing Policy:**

Affirmed.

**Final Justification:**

Thank you for the clarifying responses and additional experiments. I have raised the overall score of the paper and recommend it be accepted.

**Key Questions For Authors:**

1. The benchmark is grounded in Turiel's Domain Theory, but there are many competing frameworks in moral psychology. Is there a particular reason this taxonomy was selected over alternatives, such as Moral Foundations Theory?

2. The paper motivates the use of real-world images partly by noting the poor visual quality of AI-generated content. However, are there deeper conceptual reasons why AI-generated images are unsuitable for evaluating multimodal moral reasoning?

3. It would strengthen MORALISE's validity to include some measure of robustness or consistency. For example, augmenting images in morally trivial ways, such as cropping and testing whether model judgments remain stable. This would help establish whether models are responding to genuine moral content or to incidental visual features.

**Limitations:**

Yes

**Strengths And Weaknesses:**

The use of Turiel’s Domain Theory was interesting and the many models and analyses made for a comprehensive set of findings using MORALISE. In addition, the clarity in the presentation of findings/takeaways was appreciated.

I have some concerns/questions.

The majority-vote filtering protocol helps ensure annotation reliability but it would be valuable to release the MORALISE dataset including items that were filtered out, alongside inter-rater reliability scores. Inconsistent items among annotators may reflect genuinely morally ambiguous scenarios.
The paper should provide a demographic profile of the annotators. Given substantial evidence that university students (and ML graduate students) hold attitudes and perspectives that may not reflect the broader population, this information would be important for contextualizing the benchmark and understanding whose moral intuitions it reflects.
The paper does not specify when the proprietary model APIs were accessed. Given emerging concerns in the literature about model versioning and shadow updates to production APIs, this information is important for reproducibility.
Comparison between closed-source and open-source models relies entirely on three OpenAI models, two of which share a training lineage given that GPT-4o-mini is a distillation of GPT-4o. Broader coverage of proprietary models from other providers would be needed before drawing general conclusions about the relative moral alignment of closed versus open-source systems.

---

> ### Author Rebuttal · Authors · 2026-03-31
>
> ## W1. Ambiguous Cases
> We appreciate your interest in ambiguous cases. However, due to our decentralized annotation pipeline, filtered samples were not systematically preserved. Specifically, (1) annotators independently collected examples from the web with an explicit preference for clear moral violations rather than borderline cases, and (2) after collection, all assigned samples were labeled by multiple annotators, and those with agreement below 83% were deleted by an automatic program. **While this process is effective in ensuring a high-quality and unambiguous benchmark, it also makes it difficult to retain and release the filtered-out items.** We will clarify this limitation in the revised version.
>
> ## W2. Annotator Demographics
> Thanks for the suggestion! **We will include a more detailed demographic profile of the annotators in the revised version (subject to annotator consent)** and discuss the potential limitations and scope of the benchmark in this regard.
>
> We made deliberate efforts to ensure diversity in our annotation process: our annotators come from multiple regions, including Asia (China, Vietnam, Korea), Europe (Germany), and North America (United States). We aim for the benchmark to reflect a relatively broad set of perspectives and to mitigate strong demographic biases. In addition, to reduce potential bias arising from the annotator team, we maintained frequent group discussions during the annotation process to surface and resolve disagreements and increase awareness of cultural and subjective differences.
>
> ## W3. API Versioning and Reproducibility
> We use fixed-version proprietary models, specifically GPT-4o-2024-11-20, GPT-4o-mini-2024-07-18, and o4-mini-2025-04-16. We will explicitly mention these versions in the revised paper.
>
> ## W4. Limited Proprietary Model Coverage
> We mainly evaluated GPT models because our institution only provides reimbursement to OpenAI API expenses. Given the relatively large scale of our benchmark (over 2,400 image–text pairs), extending evaluation to additional proprietary models incurs non-trivial cost. Nonetheless, we conduct additional experiments with Gemini-3-Flash at our own expense to further validate our findings.
>
> Briefly, **Gemini achieves 80.17% accuracy on text-centric moral-norm attribution and 67.37% on image-centric attribution, which are highly consistent with the trends observed for OpenAI models.** (Detailed results will be included in the revised version). The results also well align with our main conclusions (e.g., Takeaways #1,2,3,4), suggesting that our findings are not tied to a single provider.
>
> ## Q1. Choice of Moral Taxonomy
> 1. We chose Turiel’s Domain Theory because **it provides a hierarchical and fine-grained moral taxonomy that allows us to have operational distinctions between different types of norm violations**. For example, fairness and cheating are grouped as one foundation in MFT. However, we believe that the moral issues of unfairness stemming from social prejudice and improper individual conduct (e.g., cheating) are not entirely the same and should therefore be distinguished. Thus, the latter falls under personal-integrity rather than fairness in Moralise's taxonomy.
> 2. We agree that alternative frameworks (such as MFT) are influential and valuable. Our choice should therefore not be interpreted as a claim that Turiel’s taxonomy is uniquely correct, but rather that it serves as a practical and theoretically grounded starting point for a benchmark with consistent annotation criteria. We will clarify this motivation in the revised version and discuss its relationship to alternative frameworks.
>
> ## Q2. Advantages over AI-generated Images
> Due to space limit, please kindly refer to W1 of Reviewer N84j, where we discuss the problems of AI-generated images based on recent literature, and also our position on why a non-AI-generated benchmark is a necessary complement.
>
> ## Q3. Robustness to Trivial Perturbations
> We thanks for the valuable suggestion. As suggested, we conducted a robustness analysis by applying random cropping (crop ratio = 0.8) to all images and re-evaluating performance on GPT-4o.
>
> The results show that, after perturbation, performance on text-centric tasks increases (from 69.53% to 73.01% on average), while performance on image-centric tasks decreases (from 63.25% to 59.82% on average). These findings support two key observations:
> 1. The dataset is well-curated such that, in modality-centric settings, the relevant modality carries the essential moral signal, while the other modality may introduce noise or distraction.
> 2. The overall performance change remains within a relatively small range (~3%), indicating that model behavior is reasonably stable under such perturbations and does not collapse due to minor visual changes. We will include this robustness analysis in the revised version to strengthen the validity of our conclusions.
>
> **We appreciate your helpful feedbacks and welcome further discussion.**

---

> > ### Author Rebuttal · Reviewer_UKrj · 2026-04-04
> >
> > Thank you for your response. My concerns are addressed.

---

### Official Review · Reviewer_Tezq · 2026-03-13

**Soundness:** 1
**Presentation:** 4
**Significance:** 2
**Originality:** 3
**Overall Recommendation:** 4
**Confidence:** 3

**Summary:**

This paper introduces MORALISE, a benchmark for evaluating moral alignment in vision-language models using real-world image-text pairs. The benchmark is organized around a structured moral taxonomy and includes two tasks, namely moral judgment and moral norm attribution. The paper evaluates a broad set of contemporary VLMs and highlights substantial performance gaps, especially for cases where the relevant moral signal is primarily visual.

Overall, I find the problem important and timely. A benchmark targeting moral understanding in multimodal systems is valuable, and the use of real-world image-text data is a meaningful step beyond prior text-only or synthetic-image settings. That said, I think the current paper would be stronger if it were more careful about the scope of its claims and if it provided stronger validation for its modality-based conclusions.

**Compliance With Llm Reviewing Policy:**

Affirmed.

**Key Questions For Authors:**

Can the authors add missing-modality / masking ablations to verify that the text-centric and image-centric distinctions are reflected in actual model behavior rather than only in dataset annotation?
Can the authors report more detailed evidence on annotation reliability, ideally including topic-level agreement or boundary/confusion analysis for nearby categories?
How do the authors distinguish between broad moral alignment and the narrower capability of recognizing curated moral violations and attributing associated norms?
Please clarify the benchmark’s data governance policy, especially regarding privacy-sensitive images, copyright/licensing status, and public release or redistribution.

**Limitations:**

The paper makes a useful benchmark contribution, but the current evidence more directly supports evaluation of structured moral violation recognition in multimodal settings than broad moral alignment as a general capability. The paper would be significantly stronger with clearer scope calibration, stronger modality ablations, and more detailed reliability and governance reporting.

**Strengths And Weaknesses:**

Strength:

The paper studies an important and relatively underexplored problem: moral evaluation in vision-language models rather than in text-only systems.
The use of real-world image-text pairs is a clear strength and makes the benchmark more practically relevant than settings based primarily on synthetic or AI-generated content.
The benchmark design is reasonably structured, with a nontrivial taxonomy and two task formulations that go beyond a single binary classification setting.
The empirical coverage is broad. Evaluating a large number of open- and closed-source VLMs increases the utility of the benchmark for the community.
The paper is generally clear and easy to follow, and the motivation is well articulated.
Weaknesses:
a) The paper occasionally makes claims about moral alignment in a broad sense, but the benchmark appears to measure a narrower capability: recognition and attribution of relatively structured moral violations in curated scenarios. This is still a useful contribution, but the scope of the claims should be calibrated more carefully.

b) The modality-centric conclusions are interesting, but they are not yet fully supported by controlled ablations. In particular, labeling examples as text-centric or image-centric is not the same as showing that the model’s prediction actually depends on the intended modality. Stronger missing-modality or counterfactual ablations would make these conclusions much more convincing.

c) The paper would benefit from stronger quantitative evidence on annotation reliability and category separability. For a benchmark paper, it is important not only to describe the curation pipeline, but also to show more explicitly that the topic labels and norm-attribution labels are robust, especially for conceptually adjacent categories.

d) Relatedly, the paper does not sufficiently analyze taxonomy overlap. Some categories are intuitively close, and a confusion analysis or boundary analysis would help clarify whether performance drops come from model failures or from residual ambiguity in the label space.

e) Because the benchmark is built from web-collected real-world images, the paper should provide a more explicit discussion of data governance, including privacy-sensitive content, licensing/copyright considerations, and the exact redistribution protocol.

---

> ### Author Rebuttal · Authors · 2026-03-31
>
> ## W1. Scope of Moral Alignment Claims
>
> We agree with the reviewer that the scope of our claims should be carefully calibrated. In the revised version, **we will explicitly refine our wording to ensure that the benchmark is clearly positioned as evaluating recognition and attribution of structured moral violations**.
>
> We also acknowledge the importance of other forms of moral alignment evaluation tasks, but including more tasks require substantial amount of additional work that is beyond the current scope of this work. We will include a discussion of broader moral alignment evaluation as a direction for future work.
>
>
> ## W2. Modality Ablations
> 1. **For the image-centric setting, we want to clarify that our setup is inherently a missing-text modality ablation, where model predictions rely solely on visual input.** Please note that we do NOT provide any image descriptions (i.e., the \<text\> field in the prompt templates is empty) and only include task instructions. Please see full prompt templates in Appendix B.1 for more details.
> 4. **For the text-centric setting, we have conducted an additional ablation by removing all images and retaining only textual information.** On the multi-norm attribution task, GPT-4o, GPT-4o-mini, and o4-mini achieve 81.34%, 76.97%, and 80.18%, respectively, which are consistent with or even higher than the original results (68.99%, 69.53%, 80.90%). This indicates that the images in these examples do not carry any useful moral signals, and hence removing images can improve performance. It is consistent with our annotation protocol where annotators explicitly filtered out images containing salient moral information.
>
>
> ## W3. Annotation Reliability and Separability
>
> 1. **Annotation Reliability**. We conducted an external human study with 8 non-author undergraduate/graduate volunteers, using exactly the same inputs and instructions as the VLMs. **Majority-vote human performance reaches 94.1% accuracy**, which is substantially higher than the best model performance. This gap indicates that the task is well-defined and that the annotations are reliable rather than ambiguous or noisy. We will include detailed study design and analysis in the revised version.
>
> 2. **Category Separability**. We agree that certain moral topics can be conceptually adjacent. To address this, we adopt two complementary strategies. (1) We formulate attribution as a multi-norm attribution task, allowing multiple labels to be selected simultaneously. This design reduces the reliance on strict mutual exclusivity and mitigates ambiguity between closely related categories. (2) During annotation, each sample is labeled by 12 annotators, and we discard samples with fewer than 10 agreements. **This filtering step effectively removes borderline or ambiguous cases**, ensuring high label consistency and improving separability in the final benchmark.
>
> ## W4. Taxonomy Overlap and Ambiguity
>
> As discussed in W3, our annotation pipeline is explicitly designed to minimize overlap between taxonomy categories through strict agreement filtering and careful data curation. To further validate this empirically, following the reviewer’s suggestion, we conducted a pairwise overlap analysis across all moral topic pairs in MORALISE.
>
> Concretely, for two samples from different topics, we define them as overlapping if either the image or the text modality is identical. Under this criterion, **96.15% of topic pairs exhibit zero overlap, i.e., no shared samples at all.** Among the overlapped pairs, **the maximum overlap ratio is only 1.58%.** These results provide strong quantitative evidence that the dataset has minimal taxonomy overlap, indicating that performance differences are unlikely to be driven by label-space ambiguity. Instead, they more faithfully reflect differences in model capabilities. We will include this analysis in the revised version to further clarify category separability.
>
> ## W5. Data Governance and Redistribution
> We agree that a clearer discussion of data governance is essential for a benchmark built from web-collected images, and we will explicitly document our protocol in the revision.
>
> 1. **Privacy**. We avoid collecting or including images that contain identifiable personal information or sensitive attributes whenever possible, and follow standard research practices to exclude high-risk content.
> 2. **Licensing and copyright**. All images are sourced from publicly accessible content. Since the dataset is manually curated by annotators (rather than collected via automated crawling), we explicitly screen and avoid images with potential copyright risks during collection, and will provide attribution where required.
> 3. **Redistribution**. we will clearly specify the release format and usage constraints to ensure compliance with data governance standards (e.g., providing metadata and links when appropriate).
>
> **Hope our response has addressed your concerns and welcome continued discussion.**

---

> > ### Author Rebuttal · Reviewer_Tezq · 2026-04-04
> >
> > I appreciate the authors' detailed reply and would like to see these clarifications incorporated into the revised version. I will maintain my current score.

---

### Decision · Program_Chairs · 2026-04-30

**Decision:**

Accept (regular)

**Comment:**

This paper introduces MORALISE, a benchmark for moral understanding in VLMs that uses real-world image-text data, in which models are evaluated on their ability to recognize moral violations and attribute them to appropriate social norms.

Main concerns were novelty relative to MoralBench, annotator demographics, and benchmark validation. These were largely addressed in rebuttal through additional human evaluation, stronger annotation analysis, clearer governance details, and better positioning of the benchmark as complementary to prior work. The reviewer discussion converged positively, including an increase in score from the initially most skeptical reviewer.